# Efficacy of Tocopherol vs. Chlorhexidine in the Management of Oral Biopsy Site: A Randomized Clinical Trial

**DOI:** 10.3390/jcm14030788

**Published:** 2025-01-25

**Authors:** Arianna Baldin, Clara Nucibella, Claudia Manera, Christian Bacci

**Affiliations:** Unit of Oral Pathology and Medicine and Odontostomatological Diagnostics, Section of Clinical Dentistry, Department of Neurosciences, University of Padova, 35122 Padova, Italy; arianna.baldin@studenti.unipd.it (A.B.); clara.nucibella@studenti.unipd.it (C.N.); claudia.manera@studenti.unipd.it (C.M.)

**Keywords:** biopsy, chlorhexidine, oral cavity, pain, tocopherol, wound healing

## Abstract

**Background/Objectives**: Chlorhexidine digluconate (CHX) is widely regarded as the gold standard for oral mucosa antiseptic treatments but has been associated with delayed healing, scar formation, microbiome alterations, and fibroblast toxicity. Tocopherol, with its ability to accelerate tissue healing and minimal side effects, has emerged as a potential alternative. This randomized clinical trial aimed to compare the efficacy of topical tocopherol acetate and 0.2% chlorhexidine in managing postoperative pain and wound healing following oral cavity biopsies. **Methods**: Seventy-seven patients undergoing oral biopsies were divided into two groups: the test group (tocopherol acetate) and the control group (0.2% chlorhexidine). Pain was assessed using VAS (Visual Analogue Scale) scores on days 1 and 6 postoperatively, and wound healing was evaluated through measurements of the biopsy site’s height and width from standardized photographs analyzed with ImageJ. Painkiller use was also documented. The study followed CONSORT (Consolidated Standards of Reporting Trials) guidelines, with ethical approval from the Padua Ethics Committee and registration on ISRCTN. **Results**: No significant differences were found between the groups in VAS scores, wound dimensions, or painkiller use (*p* > 0.05). However, significant pain reduction within each group was observed (*p* < 0.0001). **Conclusions**: Tocopherol acetate showed comparable efficacy to chlorhexidine, suggesting it could be a viable alternative for postoperative care in oral surgery.

## 1. Introduction

A biopsy, defined as the observation of tissue fragments taken from a living organism, is a fundamental procedure for the differential diagnosis of oral lesions, many of which are benign; however, some may be precancerous or malignant [1]. The diagnostic challenge often lies in distinguishing benign conditions from those with malignant potential, especially given that some precancerous lesions may be present in mucosae that appear clinically normal [2]. Early and accurate diagnosis is crucial to enable timely and effective therapeutic interventions.

Biopsies can be categorized into two main types: incisional biopsies, performed when a definitive histopathological diagnosis is required, and excisional biopsies, chosen for clinically benign lesions that can be entirely removed during the procedure [3]. The biopsy process is critical for constructing an etiopathogenetic framework, achieving a differential diagnosis, and planning appropriate treatment [3,4,5].

In managing biopsy sites, chlorhexidine digluconate has long been considered the gold standard due to its broad-spectrum antibacterial properties. It is commonly used in concentrations of 0.12% or 0.2% and is effective in reducing bacterial load in surgical areas. However, its prolonged use has been associated with several adverse effects, including cytotoxicity to fibroblasts, delayed wound healing, alterations in the oral microbiome, and side effects such as staining and taste alteration [3,6,7,8,9,10,11,12]. Cytogenetic studies have reported conflicting results on its impact on cell proliferation and apoptosis, and the concentration and context of its use significantly influence its effects [13,14,15,16,17,18,19,20].

Tocopherol acetate, a form of vitamin E, has emerged as a potential alternative due to its antioxidant and anti-inflammatory properties. Tocopherol plays a critical role in modulating cell membrane properties and signaling pathways, protecting polyunsaturated fatty acids, and reducing oxidative stress. It has demonstrated the ability to accelerate tissue healing by enhancing cell proliferation and collagen synthesis while exhibiting minimal side effects [21,22,23,24,25,26,27]. Unlike chlorhexidine, tocopherol is already used successfully on various mucosae, including nasal, genital, and oral tissues, with reports of good tolerability and efficacy [28,29,30,31,32,33,34,35,36,37].

The purpose of this randomized clinical trial is to evaluate the efficacy of topical tocopherol acetate compared to 0.2% chlorhexidine in managing postoperative pain and the quality of surgical wound healing in patients undergoing oral cavity biopsy.

The null hypothesis is that there are no differences between the two groups in terms of pain variation and wound healing quality.

## 2. Materials and Methods

### 2.1. Trial Design

This single-center, simple randomized, non-blinded, controlled, parallel-group study was designed according to the updated CONSORT (Consolidated Standards of Reporting Trials) 2010 guidelines for reporting parallel group randomized trials [38]. The study was conducted at the Dental Clinic of the Padua University Hospital. A total of 90 patients scheduled for oral mucosal biopsy were enrolled between January 2022 and July 2024.

Initially, a control group with no treatment was chosen. Since chlorhexidine treatment is the gold standard and the most commonly prescribed treatment, it was decided to assign chlorhexidine treatment to the control group.

The methodological variation was registered on ISCTRN.

### 2.2. Participants

All surgical procedures were performed by the same examiner (A.B.). To induce local anesthesia, articaine hydrochloride 40 mg/mL with adrenaline 5 μg/mL was used. A 15 blade mounted on a handle with a 3 or 5 mm punch was used. The suture was performed with interrupted stitches using a 3.0 or 4.0 polyglycolic acid suture (Polysorb™, Covidien, Dublin, Ireland). Where suturing was not possible for anatomical reasons, regenerated oxidized cellulose was used (Tabotamp^®^, Ethicon, Raritan, NJ, USA). Photographs were taken before and at the end of the procedure with the aid of a millimeter periodontal probe to standardize and analyze the images with ImageJ version 1.54k (Wayne Rasband, US National Institute of Health, Bethesda).

For participant assignment, simple randomization was used with a coin toss by an examiner (C.B.) different from the examiner performing the surgical procedure (A.B.). The randomization list was kept in another building in a locked cabinet that could only be opened by the examiner who performed the randomization (C.B.). Another examiner (C.N.) enrolled the participants by performing the implementation. The examiner who performed the randomization (C.B.) communicated the assignment group to a fourth examiner (C.M.) at the end of the surgical procedure. Recruitment ended because the predetermined time limit was reached.

In Table 1, inclusion and exclusion criteria are shown.

Patients were given postoperative instructions as per the normal clinical practice of the clinic (no vigorous rinsing, no hot or hard foods, no ice application) and to take standard analgesic therapy (paracetamol 1000 mg every 8 h or ibuprofen 600 mg every 6 h) if necessary. They were also instructed to complete, on the first and sixth postoperative days, two sheets with an identification code with the representation of VAS-P (Visual Analogue Scale for Pain).

Patients assigned to the test group TG (tocopherol acetate) were instructed to perform three topical applications per day of tocopherol acetate (oral oily gel of 100% tocopherol acetate, Vea Filme Os, Hulka S.r.l., Rovigo, Italy) for 7 days after oral biopsy, spreading the gel on the lesion until it was covered. It was specified that it was not necessary to massage, and it was advisable to avoid touching the area even with the tongue for at least 1–2 min to promote the formation of the protective film. Patients assigned to the control group CG (chlorhexidine digluconate) were instructed to perform three topical applications per day of spray containing 0.2% chlorhexidine (Corsodyl, Haleon S.r.L., Milano (MI), Italy) for 7 days after oral biopsy.

Patients were re-evaluated by examiner C.N. on the sixth postoperative day, performing suture removal, photography with millimeter probe, collecting VAS-P recording, and noting adverse effects. The areas subjected to surgical intervention were divided into two subgroups:Attached gingiva: attached gingiva, retromolar trigone, edentulous ridge;Loose mucosa: buccal mucosa, tongue, floor of mouth, lips, retrocommissure.

### 2.3. Outcomes

The primary outcome was pain reduction from the first to the sixth postoperative day, measured on a visual analog scale, VAS-P, which corresponds to the visual representation of the amplitude of pain perceived by the patient and consists of a predetermined line 10 cm long, where the left end corresponds to no pain, and the right end to the worst possible pain. The patient is asked to mark on the line a sign that represents the level of pain experienced. At the 6-day check, VAS-P recordings (completed by the patient on the first and sixth day) were collected, and the numerical score was realized using a graduated scale placed on the back of the millimeter ruler (rounding down to 0.49 and up from 0.50).

The secondary outcome was to evaluate the quality of wound healing by measuring the height and width of the postoperative site photographed immediately after surgery and on the sixth postoperative day. The photos were analyzed and standardized with ImageJ (ImageJ 1.53t; Java 1.8.0_345 version).

The tertiary outcome was the variation of VAS-P in patients who took painkillers compared to patients who did not take them.

The outcomes were not changed during the trial.

Other factors were also considered: sex, age, smoking habit.

### 2.4. Sample Size

As there was no a priori information on the degree of difference expected to be found, a minimum number of 30 patients per group was taken with a two-year enrollment window. At the end of the enrollment period, the study was terminated with a number of 77 patients randomized at the end of implementation. The minimum number of participants had been reached, but the patients were not sufficient for a 1:1 randomization.

### 2.5. Blinding

It was not possible to conduct the trial blind for patients due to the obvious difference in formulation and consistency (gel vs. liquid) and taste of the product. The professional (A.B.) who provided the care, the subject who collected the data (C.N.), the outcome evaluator (A.B.), and the statistician (M.C.) who analyzed the data were kept blinded.

### 2.6. Statistical Methods

Numerical variables were summarized as median and interquartile range (IQR), while categorical variables as absolute and relative frequency (percentage). Changes from day 1 to day 6 in VAS and wound dimensions were compared between the two groups (tocopherol acetate vs. chlorhexidine digluconate) with the Mann–Whitney test, and the effect was calculated as the difference of medians with a 95% bootstrap confidence interval. The occurrence of complications was compared between the two groups with Fisher’s test, and the effect was calculated as a risk ratio with a 95% confidence interval. The analysis was performed according to the “intention-to-treat” approach. Since 7 patients (9%) did not follow the prescribed therapy, a “per protocol” analysis of outcomes was also performed to provide more information to the reader. As exploratory analyses, the variation of VAS stratified between those who took and those who did not take painkillers, and the variation of wound dimensions between smokers and non-smokers were investigated. Furthermore, the association between sex and painkiller intake was investigated with the chi-square test. A *p* < 0.05 was considered statistically significant. The limited sample size suggests caution in interpreting the results. Statistical analysis was performed with R 4.3 (R Foundation for Statistical Computing, Vienna, Austria).

## 3. Results

Of the total of 90 patients evaluated for eligibility by two examiners (C.N. and C.M.), 7 patients did not meet the inclusion criteria (4 underage patients, 2 patients taking immunomodulators, 1 patient taking antiangiogenics), 6 did not agree to participate in the trial, and 77 patients were randomized, of which 51 were women and 26 men (Figure 1).

The median age was 62 years (IQR 55–73); 40 patients were assigned to the test group (tocopherol acetate) and 37 to the control group (chlorhexidine digluconate). Table 2 shows the characteristics of patients in individual groups.

Of these, 5 patients in the test group and 2 patients in the control group did not take the treatment, reporting at the 6-day check that they did not consider it necessary due to absence of pain or because they had forgotten to apply the treatment; of the 5 patients in the test group, 1 patient also missed the check-up, so only pain was recorded via the VAS-P scale, which led to a subsequent check; for the other 6 patients, pain was still recorded via VAS-P at 1 and 6 days, and wound dimensions were measured.

Painkiller intake was similar between women (15/51, 29%) and men (7/26, 27%) (*p* = 0.99).

Regarding the primary outcome, the analysis of VAS variation from day 1 to day 6 is reported in Table 2, Table 3 and Table 4. VAS decreased from day 1 to day 6 both in those treated with topical tocopherol (*p* < 0.0001) and in those subjected to usual treatment (*p* < 0.0001). In general, however, the variation in VAS was not statistically different in the two treatment groups (Table 3a). The same results were confirmed using the “per protocol” approach (Table 3b).

Figure 2 shows the graphical representation of VAS variation from day 1 to day 6 in the two groups.

### 3.1. VAS Stratified Between Those Who Have and Those Who Have Not Taken Painkillers

Even when stratifying between those who took and those who did not take painkillers, the variation in VAS was not statistically different in the two treatment groups (Table 4 and Table 5).

### 3.2. Quality of Wound Healing

Regarding the secondary outcome, the analysis of wound healing quality in terms of dimensions (height H and width W) is reported in Table 5 and Table 6. In general, the variation in dimensions (height and width) was not statistically different in the two treatment groups (Table 6a and Table 7a). The analysis was not performed on patients who did not show up for follow-up, as it was not possible to collect the data. The same results were confirmed using the “per protocol” approach (Table 6b and Table 7b).

Figure 3 shows a graphical representation via Tukey box plot of the variation of H and W in the two groups, and Figure 4 compares the initial and final values of H and W in the two groups.

### 3.3. Quality of Wound Healing Stratified Between Smokers and Non-Smokers

Even when stratifying between smokers and non-smokers, the variation in dimensions was not statistically different in the two treatment groups (Table 8, Table 9, Table 10 and Table 11). The analysis was not performed on patients who did not show up for follow-up and were simultaneously smokers or non-smokers.

### 3.4. Quality of Wound Healing: Comparison Between “Attached” and “Loose”

Comparing the wound healing quality between “attached” and “loose” sites, the variation in dimensions was statistically different in the two sites (Table 12 and Table 13). In particular, a greater reduction in height was observed for the “loose” site and a greater reduction in width for the “attached” site.

### 3.5. Adverse Effects

No complications occurred in the test group (0%), whereas three complications were identified in the control group (8%). These complications were recorded as burning sensation (*n* = 2), and in one patient (*n* = 1), a submucosal effusion on the external skin at the cheek level was observed, which at the 6-day follow-up was already in the healing phase. The difference was not statistically significant (Table 14a). The same results were confirmed using the “per protocol” approach (Table 14b).

## 4. Discussion

The results of this trial have demonstrated a non-significant difference between the two treatments compared, tocopherol acetate and chlorhexidine digluconate 0.2%, in alleviating pain at the oral biopsy surgical site in the postoperative period (*p*-value 0.99). Thus, the null hypothesis was confirmed. The maximum pain recorded at day 1 postoperative was a value of 10 in two patients (*n* = 1 patient in the test group who had the lip as the surgical site and *n* = 1 patient in the control group who had the palate as the surgical site); the maximum pain recorded on the sixth day was 8 (*n* = 1 patient in the test group who had the attached gingiva as the surgical site). Unexpectedly, the use of painkillers did not improve postoperative pain, resulting statistically not different between those who took them and those who did not. It could also be hypothesized that some patients take analgesic therapy to prevent postoperative pain and not effectively as treatment. It is interesting to note how pain was perceived equally by women and men. It should be emphasized that in both groups, pain decreased significantly.

To analyze the photos taken immediately after the procedure and at 6 days, the ImageJ software version 1.54k was used. This program standardizes images based on a known length measurement and has been used by several authors to analyze digital images [34,39,40,41]. The photos were all taken with a millimeter periodontal probe before the intervention, after the intervention, and at 6 days. The photos used were those taken after the intervention and at 6 days. To try to minimize error in the evaluation of wound healing, it was decided to compare the sites also based on the anatomy of the region where the biopsy was performed, namely in loose mucosa (buccal mucosa, tongue, floor of mouth, lips, retrocommissure) (Figure 5) or attached (vestibular attached gingiva, retromolar trigone, edentulous ridge) (Figure 6) for their different extensibility.

Figure 7, Figure 8, Figure 9, Figure 10, Figure 11 and Figure 12 show the process of analyzing photos through ImageJ version 1.54k.

In the present study, however, no significant differences were recorded in the variation of wound size in terms of height and width between the two treatments. Statistically significant differences emerged in the measurement of the variables ΔH and ΔW referring to sites differentiated into loose mucosa and attached mucosa. This result finds explanation in the fact that lesions in loose mucosa were all sutured with one or more interrupted stitches, so they appeared in the postoperative period wide in height (5.6; 8.0), but with edges approximated in width (1.1; 2.2); thus at 6 days, the decrease in height was statistically significant (2.3; 5.2), while a small solution of continuity remained at the suture level (1.1; 1.8). Therefore, the width does not vary. This does not happen for the attached mucosa, where the height does not vary significantly, as the lesion still remains wide looking at the before (5.1; 6.8) and after (2.4; 5.2), with the edges not being approximated. The variation is only 1.4. The width decreases significantly compared to the “loose mucosa” site, but overall, it is still wider: 5.5 mm (2.8; 7.1) in the postoperative period and 2.6 mm (1.3; 4.8) at 6 days for attached mucosa compared to 1.5 mm (1.1; 2.2) and 1.3 (1.1; 1.8), respectively, for loose mucosa.

So overall, loose mucosa seems to heal better than attached mucosa.

This allows us to observe, however, that tocopherol treatment is not worse than chlorhexidine treatment, which represents the gold standard.

For this study, it was chosen to treat the test group with tocopherol and the control group with chlorhexidine. After choosing “no treatment” as the control group, it was decided to change the treatment using chlorhexidine, which represents the gold standard [3,42], as it is usually used after oral surgery.

There is no consensus in the literature regarding the effects of chlorhexidine in the oral cavity. While some studies claim that it causes irritation, pigmentation, taste alteration, and oral microbiome alteration in addition to reduced cell proliferation, having toxic effects on fibroblasts [8,9,10,11,13,20], others state that it does not lead to cellular alteration or cytogenetic damage, nor does it cause a negative effect on wound healing [7,14,17]. However, it is considered a gold standard drug in the antiseptic treatment of oral mucosae [3].

Tocopherol acetate has long been widely used on the skin [29,32] and more recently also on mucosae [28,30,34]. In this study, it was decided to use tocopherol due to the absence of side effects described in the literature [23,35,36,37], and also due to the formulation of the product used, which has acetate oil as the main phase, having no excipients and thus fewer triggers for the appearance of side effects [31], and for the benefits described both in vitro [27] and in vivo [25].

The biopsy procedure was chosen as the surgical procedure to compare tocopherol acetate and chlorhexidine because it is a standard procedure with comparable results, usually performable in a short time and which does not cause severe pain in most cases, as might be the case with third molar surgery.

Unexpectedly, there are few studies in the literature, insufficiently in-depth, about the use of a vasoconstrictor in the biopsy procedure. The study by Margarone et al. [43] in 1985 histologically investigated various artifacts created during the biopsy sampling. Here, the use of 2% lidocaine with 1:100,000 epinephrine is mentioned, reporting that if hemostasis is necessary, infiltration can be used deeply relative to the lesion or infiltration after biopsy sampling. The study by Avon and Klieb [44] from 2012 reports the use of exactly the same anesthetic. Our clinical experience has not shown contraindications or artifact formation in the use of a vasoconstrictor, particularly adrenaline, in low concentrations (5 micrograms/mL), associated with local anesthetic to obtain an anesthetic effect in view of performing biopsies of oral soft tissues. It could be clinically advantageous as it reduces intraoperative bleeding, improving visibility.

Contrary to what one might think, there are no indications and contraindications in the literature about the possibility of performing suturing at the biopsy site and the problems associated with it (for example, territorial or hematic dissemination in the case of suturing in a tissue with an unexcised lesion). In the study by Jeng et al. [45] and Avon and Klieb [44], the possibility of performing suturing after the biopsy procedure is reported without further elaboration.

Regarding tobacco consumption, in this trial, 64 patients (83%) were non-smokers and 13 smokers (17%). It is widely documented in the literature that cigarette smoking represents a significant obstacle to wound healing. Tobacco, particularly nicotine, acts as a vasoconstrictor, causing a reduction in blood flow to the tissues involved in surgery, also due to its catecholamine release power and platelet aggregation, determining a risk of microvascular occlusion. The combination of vasoconstriction and microthrombosis leads to a condition of tissue ischemia [46]. Despite this, in this study, no significant difference in healing was evidenced between the group of smokers and the group of non-smokers; this result could be due to the limited sample size.

Having emerged that there is no difference between the two treatments, the use of tocopherol acetate could be considered as an alternative to chlorhexidine in the treatment of the post-surgical site in the oral cavity, given the absence of side effects, which instead occur in chlorhexidine treatment (pigmentation and microbiome alteration in particular). It remains, however, fundamental for surgical wounds that require antiseptic treatment.

This study has several limitations that should be considered. First, it was not possible to conduct the study in a double-blind manner due to the evident differences in consistency and application methods of the treatments (gel vs. spray). Additionally, the sample size of 77 patients, while sufficient to detect primary differences, can be considered relatively small, limiting the generalizability of the results. Another limitation lies in the follow-up period, which focused on the first six postoperative days; long-term monitoring could provide more comprehensive information on the effects of the treatments on wound healing. Moreover, while height and width of the lesion were measured to assess wound healing, including the area of the lesion would have provided more robust and detailed data on the healing process. Lastly, the absence of a placebo group or a no-treatment group limits the ability to evaluate the absolute efficacy of each intervention.

Further studies with a larger sample size, with a longer follow-up, and considering other types of surgical interventions such as post-extraction sites are deemed necessary. Future studies with a larger sample could focus on the effectiveness of the two treatments on wound healing quality by identifying four different study groups based on treatment and biopsy site (loose, attached). Another study arm with no treatment and one with a placebo could have been used to increase the validity of the study, but this would imply a larger sample.

## 5. Conclusions

The results that emerged from this study indicate that there are no statistically significant differences between the application of tocopherol acetate or chlorhexidine digluconate 0.2% at the oral biopsy site in terms of decrease in perceived pain and wound healing quality; given the absence of side effects, which instead occur in chlorhexidine treatment, the use of tocopherol acetate could be considered as an alternative to chlorhexidine in the treatment of the post-surgical site in the oral cavity. Further studies are deemed necessary to confirm the results obtained.

## Figures and Tables

**Figure 1 jcm-14-00788-f001:**
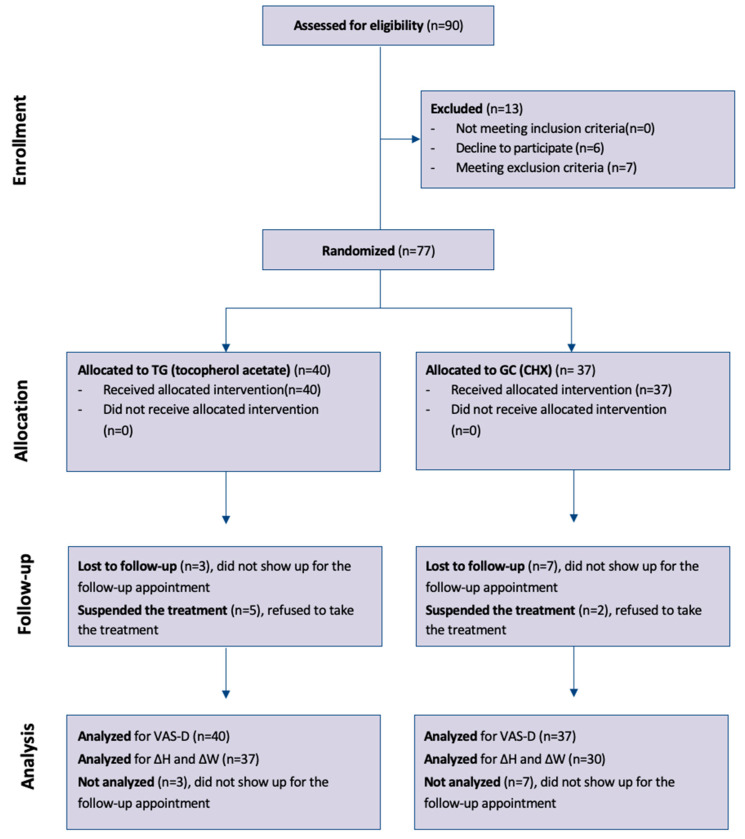
Participant flowchart according to CONSORT diagram [38]. ΔH and ΔW: Changes from day 1 to day 6 in wound dimensions were compared between the two groups (tocopherol acetate vs. chlorhexidine digluconate) with the Mann–Whitney test, and the effect was calculated as the difference of medians with 95% bootstrap confidence interval.

**Figure 2 jcm-14-00788-f002:**
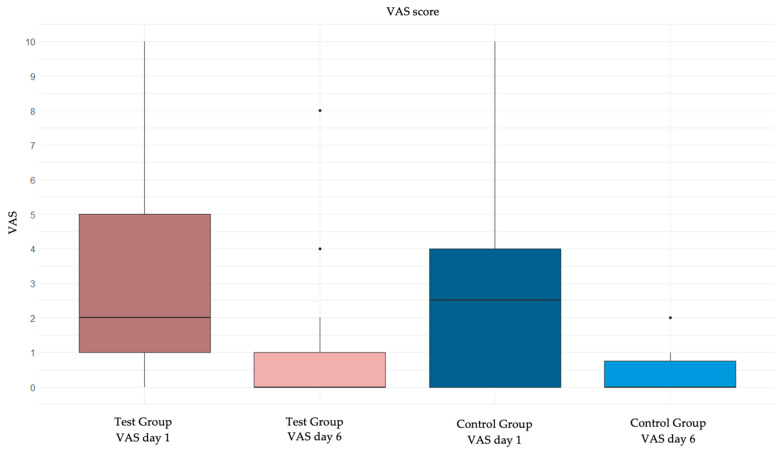
Box plot of the change in VAS from day 1 to day 6 in the two groups.

**Figure 3 jcm-14-00788-f003:**
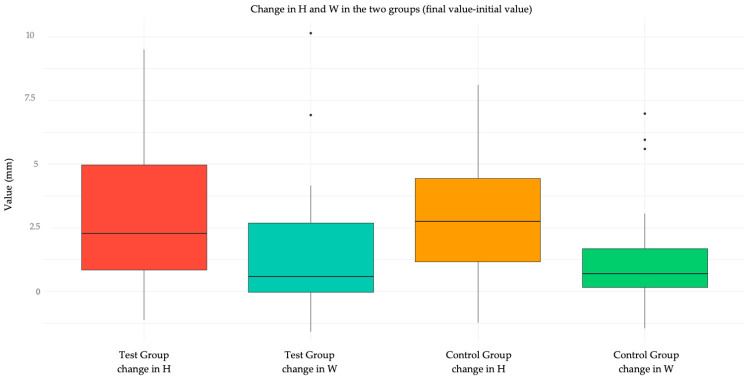
Box plot showing the change in H and W in the test group and in the control group.

**Figure 4 jcm-14-00788-f004:**
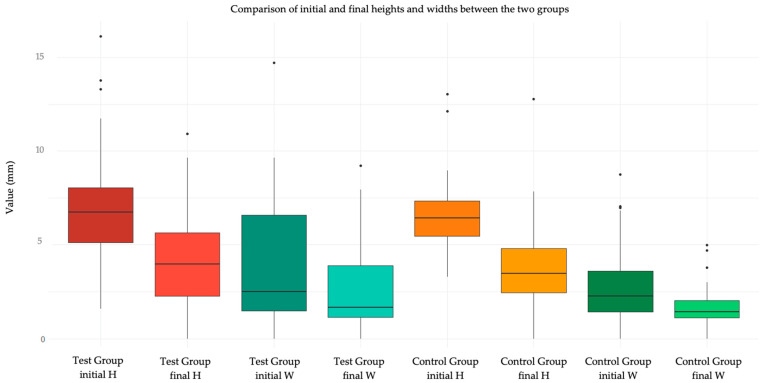
Box plot showing the initial and final values of H and W in test group and control group.

**Figure 5 jcm-14-00788-f005:**
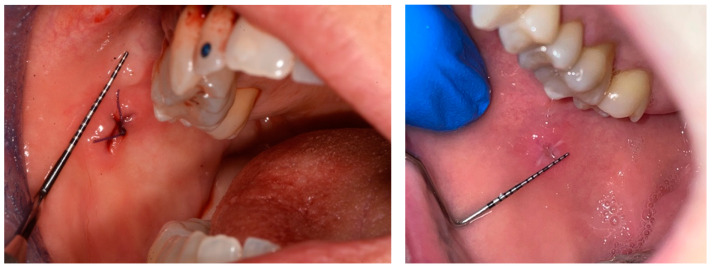
A biopsy in the loose mucosa (buccal mucosa) in the immediately postop and at day 6.

**Figure 6 jcm-14-00788-f006:**
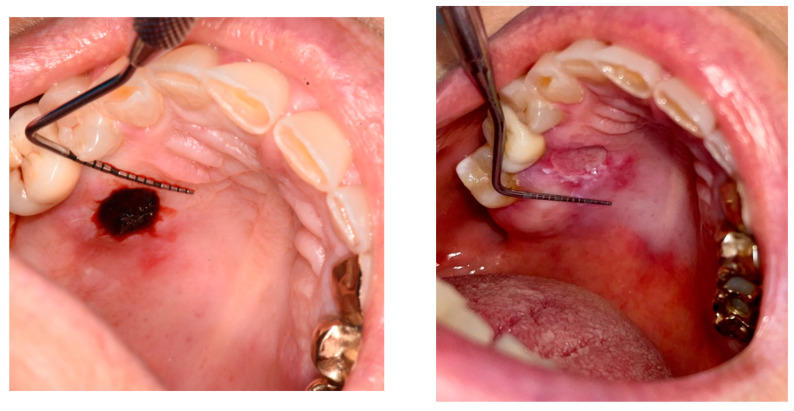
A biopsy in the attached mucosa (palate) in the immediately postop and at day 6.

**Figure 7 jcm-14-00788-f007:**
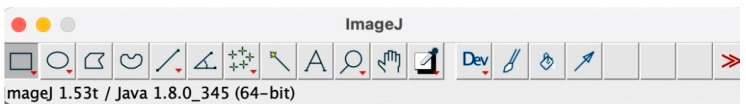
ImageJ splash screen when opening the program.

**Figure 8 jcm-14-00788-f008:**
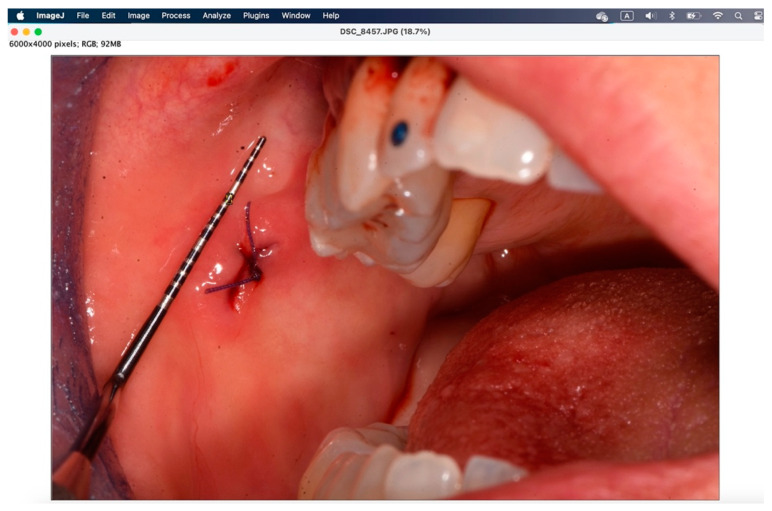
With the “straight” instrument, the fifth starting from the right, the measurement of 1 mm is drawn on the periodontal probe.

**Figure 9 jcm-14-00788-f009:**
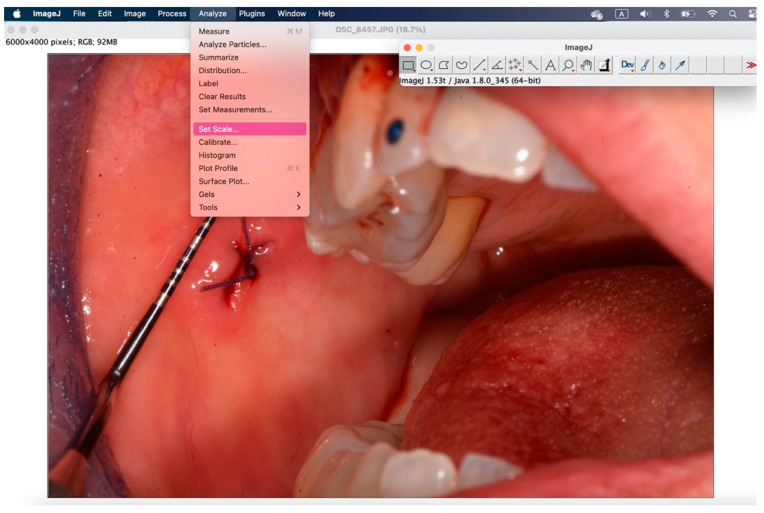
Using “analyze” > “set scale” you configure the known length.

**Figure 10 jcm-14-00788-f010:**
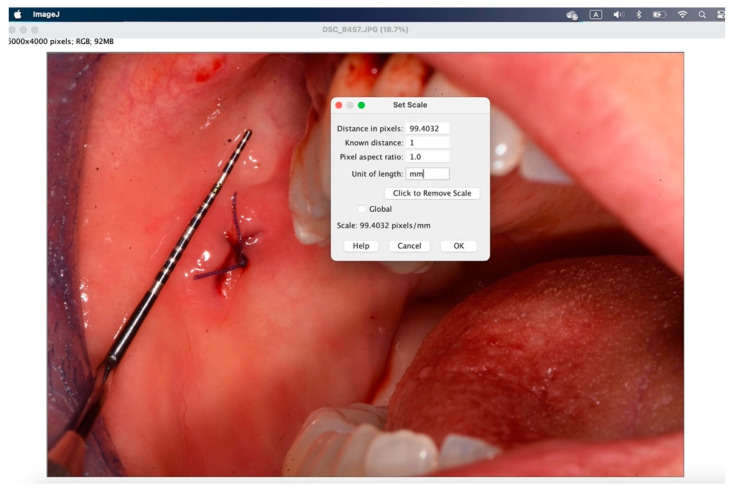
In the window that appears, enter the “known distance” and the “unit of length”, then 1 mm.

**Figure 11 jcm-14-00788-f011:**
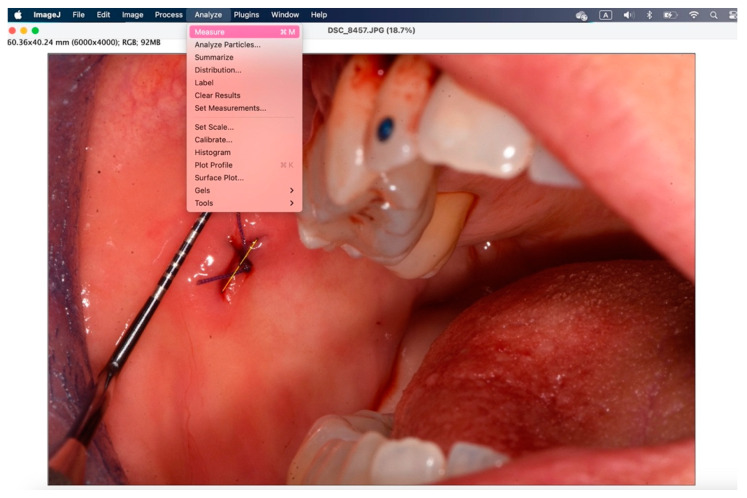
With the “straight” instrument you go to trace the measurement on the lesion, and through “analyze” > “measure” you obtain the desired measurement in millimeters.

**Figure 12 jcm-14-00788-f012:**
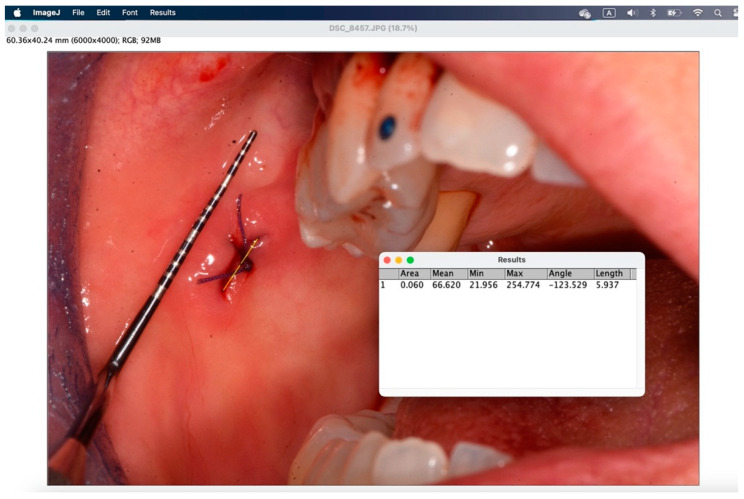
Through the window that appears, the measurement taken in the set unit of measurement is displayed in “length”.

**Table 1 jcm-14-00788-t001:** Inclusion and exclusion criteria.

Inclusion Criteria	Exclusion Criteria
Patients scheduled for oral cavity biopsy at the University of Padua Dental Clinic Patients able to read and understand the information sheet and express informed consent Patients who, after being informed in detail about the treatment usage methods, adhered to the administration scheme	Underage patients Patients on antiangiogenic therapy Patients on immunomodulator therapy Patients with known hypersensitivity to the device

**Table 2 jcm-14-00788-t002:** Characteristics of the patients.

	Test Group (*n* = 40)	Control Group (*n* = 37)
sex, *n* (%):	27 (68%)	24 (65%)
women	13 (32%)	13 (35%)
men	27 (68%)	24 (65%)
age, years: median (IQR)	64 (54–74)	61 (56–70)
mucosal site, *n* (%):		
attached	18 (45%)	18 (49%)
loose	22 (55%)	19 (51%)
painkillers, *n* (%)	14 (35%)	8 (22%)
smoker, *n* (%):		
no	26 (65%)	26 (70%)
yes	9 (23%)	4 (11%)
ex	5 (12%)	7 (19%)

**Table 3 jcm-14-00788-t003:** (**a**) Change in VAS from day 1 to day 6. (**b**) Change in VAS from day 1 to day 6 (“per protocol approach”).

**(a)**				
	**Test Group** **(*n* = 40)**	**Control Group (*n* = 37)**	**Median Difference (95% Bootstrap Confidence Interval)**	***p* Value**
VAS day 1: median (IQR)	2 (1; 4)	2 (0; 4)		
VAS day 6: median (IQR)	0 (0; 1)	0 (0; 0)		
Variation in VAS: median (IQR)	2 (1; 3)	2 (0; 4)	0 (from −1 to 1)	0.99
**(b)**				
	**Test Group** **(*n* = 35)**	**Control Group (*n* = 35)**	**Median Difference (95% Bootstrap Confidence Interval)**	***p* Value**
VAS day 1: median (IQR)	2 (1; 4)	2 (0; 4)		
VAS day 6: median (IQR)	0 (0; 1)	0 (0; 1)		
Variation in VAS: median (IQR)	2 (1; 3)	2 (0; 4)	0 (from −1 to 1)	0.99

VAS: Visual Analogue Scale. IQR: Interquartile Range.

**Table 4 jcm-14-00788-t004:** Change in VAS from day 1 to day 6 in those who have not taken painkillers.

	Test Group (*n* = 26)	Control Group (*n* = 29)	Median Difference (95% Bootstrap Confidence Interval)	*p* Value
VAS day 1: median (IQR)	2 (0; 4)	2 (0; 4)		
VAS day 6: median (IQR)	0 (0; 0)	0 (0; 0)		
Variation in VAS: median (IQR)	2 (0; 3)	2 (0; 3)	0 (from −1 to 2)	0.83

VAS: Visual Analogue Scale. IQR: Interquartile Range.

**Table 5 jcm-14-00788-t005:** Change in VAS from day 1 to day 6 in those who have taken painkillers.

	Test Group (*n* = 14)	Control Group (*n* = 8)	Median Difference (95% Bootstrap Confidence Interval)	*p* Value
VAS day 1: median (IQR)	5 (2; 8)	4 (2; 5)		
VAS day 6: median (IQR)	0 (0; 3)	0 (0; 1)		
Variation in VAS: median (IQR)	2 (1; 6)	2 (2; 4)	0 (from −4 to 3)	0.92

VAS: Visual Analogue Scale. IQR: Interquartile Range.

**Table 6 jcm-14-00788-t006:** (**a**) Change in height from day 1 to day 6 (information not available in 10 patients). (**b**) Change in height from day 1 to day 6 (“per protocol” approach).

**(a)**				
	**Test Group** **(*n* = 37)**	**Control Group (*n* = 30)**	**Median Difference (95% Bootstrap Confidence Interval)**	***p* Value**
Height at day 1: median (IQR)	6.7 (5.1; 8.0)	6.4 (5.4; 7.3)		
Height at day 6: median (IQR)	4.0 (2.3; 5.6)	3.5 (2.4; 4.8)		
Variation in height: median (IQR)	2.2 (0.8; 4.9)	2.7 (1.2; 4.1)	−0.5 (from −1.6 to 1.7)	0.91
**(b)**				
	**Test Group** **(*n* = 33)**	**Control Group (*n* = 28)**	**Median Difference (95% Bootstrap Confidence Interval)**	***p* Value**
Height at day 1: median (IQR)	6.7 (5.1; 8.1)	6.4 (5.4; 7.3)		
Height at day 6: median (IQR)	4.2 (2.3; 6.0)	3.5 (2.4; 4.8)		
Variation in height: median (IQR)	2.0 (0.8; 4.9)	2.5 (1.0; 4.2)	−0.5 (from −2.4 to 1.0)	0.85

IQR: Interquartile Range.

**Table 7 jcm-14-00788-t007:** (**a**) Change in width from day 1 to day 6 (information not available in 10 patients). (**b**) Change in width from day 1 to day 6 (“per protocol” approach).

**(a)**				
	**Test Group** **(*n* = 37)**	**Control Group (*n* = 30)**	**Median Difference (95% Bootstrap Confidence Interval)**	***p* Value**
Width at day 1: median (IQR)	2.5 (1.5; 6.6)	2.2 (1.4; 3.5)		
Width at day 6: median (IQR)	1.7 (1.1; 3.9)	1.4 (1.1; 2.0)		
Variation in width: median (IQR)	0.4 (−0.1; 2.7)	0.7 (0.2; 1.5)	−0.2 (from −0.9 to 0.6)	0.71
**(b)**				
	**Test Group** **(*n* = 33)**	**Control Group (*n* = 28)**	**Median Difference (95% Bootstrap Confidence Interval)**	***p* Value**
Width at day 1: median (IQR)	2.5 (1.5; 6.5)	2.4 (1.4; 3.9)		
Width at day 6: median (IQR)	1.7 (1.1; 3.9)	1.5 (1.1; 2.2)		
Variation in width: median (IQR)	0.5 (−0.2; 1.6)	0.7 (0.2; 1.5)	−0.2 (from −1.1 to 0.5)	0.56

IQR: Interquartile Range.

**Table 8 jcm-14-00788-t008:** Change in height from day 1 to day 6: non-smokers (information not available in 8 patients).

	Test Group (*n* = 24)	Control Group (*n* = 20)	Median Difference (95% Bootstrap Confidence Interval)	*p* Value
Height at day 1: median (IQR)	6.6 (5.1; 7.3)	6.4 (5.4; 7.3)		
Height at day 6: median (IQR)	3.1 (2.0; 5.4)	2.8 (2.4; 3.9)		
Variation in height: median (IQR)	2.4 (1.1; 4.4)	2.9 (1.3; 4.7)	−0.5 (from −2.1 to 1.6)	0.63

IQR: Interquartile Range.

**Table 9 jcm-14-00788-t009:** Change in width from day 1 to day 6: non-smokers (information not available in 8 patients).

	Test Group (*n* = 24)	Control Group (*n* = 20)	Median Difference (95% Bootstrap Confidence Interval)	*p* Value
Width at day 1: median (IQR)	2.9 (1.2; 5.8)	2.2 (1.2; 3.0)		
Width at day 6: median (IQR)	1.6 (0.9; 3.3)	1.4 (1.1; 1.7)		
Variation in width: median (IQR)	0.6 (−0.2; 2.8)	0.6 (0.1; 1.4)	0.0 (from −1.1 to 1.8)	0.86

IQR: Interquartile Range.

**Table 10 jcm-14-00788-t010:** Change in height from day 1 to day 6: smokers (information not available in 2 patients).

	Test Group (*n* = 14)	Control Group (*n* = 11)	Median Difference (95% Bootstrap Confidence Interval)	*p* Value
Height at day 1: median (IQR)	6.9 (5.6; 9.8)	6.3 (5.7; 7.1)		
Height at day 6: median (IQR)	5.2 (4.0; 6.7)	4.9 (2.5; 5.2)		
Variation in height: median (IQR)	1.7 (0.7; 5.2)	2.6 (0.9; 3.3)	−0.9 (from −2.4 to 3.8)	0.65

IQR: Interquartile Range.

**Table 11 jcm-14-00788-t011:** Change in width from day 1 to day 6: smokers (information not available in 2 patients).

	Test Group (*n* = 14)	Control Group (*n* = 11)	Median Difference (95% Bootstrap Confidence Interval)	*p* Value
Width at day 1: median (IQR)	2.5 (1.9; 6.6)	2.1 (1.6; 5.4)		
Width at day 6: median (IQR)	2.7 (1.3; 5.6)	2.0 (1.1; 3.0)		
Variation in width: median (IQR)	0.4 (0.1; 1.0)	0.9 (0.5; 1.8)	−0.5 (from −1.8 to 0.5)	0.38

IQR: Interquartile Range.

**Table 12 jcm-14-00788-t012:** Change in height from day 1 to day 6.

	Attached Gingiva (*n* = 32)	Loose Mucosa (*n* = 35)	Median Difference (95% Bootstrap Confidence Interval)	*p* Value
Height at day 1: median (IQR)	6.2 (5.1; 6.8)	6.9 (5.6; 8.0)		
Height at day 6: median (IQR)	4.1 (2.4; 5.2)	3.1 (2.3; 5.2)		
Variation in height: median (IQR)	1.4 (0.5; 3.6)	3.2 (.9; 5.1)	−1.8 (from −3.4 to −0.2)	0.008

IQR: Interquartile Range.

**Table 13 jcm-14-00788-t013:** Change in width from day 1 to day 6.

	Attached Gingiva (*n* = 32)	Loose Mucosa (*n* = 35)	Median Difference (95% Bootstrap Confidence Interval)	*p* Value
Width at day 1: median (IQR)	5.5 (2.8; 7.1)	1.5 (1.1; 2.2)		
Width at day 6: median (IQR)	2.6 (1.3; 4.8)	1.3 (1.1; 1.8)		
Variation in width: median (IQR)	1.3 (0.6; 3.2)	0.1 (−0.5; 0.7)	1.1 (from 0.6 to 2.7)	<0.0001

IQR: Interquartile Range.

**Table 14 jcm-14-00788-t014:** (**a**) Adverse effects. (**b**) Adverse effects (“per protocol” approach).

**(a)**				
	**Test Group** **(*n* = 40)**	**Control Group (*n* = 37)**	**Risk Ratio (95% Bootstrap Confidence Interval)**	***p* Value**
Adverse effects, *n* (%)	0 (0%)	3 (8%)	0.15 (from 0.01 to 2.94)	0.11
**(b)**				
	**Test Group** **(*n* = 35)**	**Control Group (*n* = 35)**	**Risk Ratio (95% Bootstrap Confidence Interval)**	***p* Value**
Adverse effects, *n* (%)	0 (0%)	3 (9%)	0.16 (from 0.01 to 3.16)	0.24

## Data Availability

The original contributions presented in this study are included in the article. Further inquiries can be directed to the corresponding author.

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
