# Peer review of "Efficacy of Tocopherol vs. Chlorhexidine in the Management of Oral Biopsy Site: A Randomized Clinical Trial"

_jcm, 2025, doi:10.3390/jcm14030788_

Round 1
Reviewer 1 Report
Comments and Suggestions for Authors
I suggest expanding in the methodology the broader description of how the healing quality was measured. What is meant by the term, quality for your study.
Indicar también en qué concentración se utilizó el acetato de tocoferol, de qué marca o de qué proveedor, así como en el caso de la clorhexidina, entrar en más detalle sobre el producto utilizado y tanto para su presentación. Se sugiere complementar la discusión si existen diferencias en la cicatrización de la herida según el tipo de biopsia de la herida que se realizó y presentar los resultados del tamaño de la herida, según lo indicado en el protocolo, para los tipos de pacientes analizados, pacientes con Encía adherida: encía adherida, trígono retromolar, cresta edéntula Mucosa suelta: mucosa bucal, lengua, suelo de la boca, labios, retrocomisura.
Se sugiere complementar y presentar los resultados del tamaño de la herida, según lo indicado en el protocolo, para los tipos de pacientes analizados, pacientes con Encía adherida: encía adherida, trígono retromolar, cresta edéntula Mucosa suelta: mucosa bucal, lengua, piso de la boca, labios, retrocomisura.

Author Response
RESPONCES REVIEWER 1
- I suggest expanding in the methodology the broader description of how the healing quality was measured. What is meant by the term, quality for your study.
Re: The entire process is described with the aid of images at line 274-305. The quality of wound healing is defined as a reduction in the height and width of the wound.
- Indicar también en qué concentración se utilizó el acetato de tocoferol, de qué marca o de qué proveedor…
Re: Thank for clarification, as requested I corrected with “oral oily gel of 100% tocopherol acetate, Vea Filme Os, Hulka S.r.l., Rovigo” at line 103-104
…así como en el caso de la clorhexidina, entrar en más detalle sobre el producto utilizado y tanto para su presentación.
Re: Thank for clarification, as requested I corrected with “Corsodyl, Haleon S.r.L., Milano (MI)” at line 109
- Se sugiere complementar la discusión si existen diferencias en la cicatrización de la herida según el tipo de biopsia de la herida que se realizó y presentar los resultados del tamaño de la herida, según lo indicado en el protocolo, para los tipos de pacientes analizados, pacientes con Encía adherida: encía adherida, trígono retromolar, cresta edéntula Mucosa suelta: mucosa bucal, lengua, suelo de la boca, labios, retrocomisura.
Re: I was not able to find any articles in the literature discussing the differences in wound healing based on the type of biopsy performed.
- Se sugiere complementar y presentar los resultados del tamaño de la herida, según lo indicado en el protocolo, para los tipos de pacientes analizados, pacientes con Encía adherida: encía adherida, trígono retromolar, cresta edéntula Mucosa suelta: mucosa bucal, lengua, piso de la boca, labios, retrocomisura.
Re: These results are already present presented as attached gingiva and loose mucosa as the sample size did not allow to create eight subgroups.
The paper has been thoroughly revised, and the changes have been highlighted in yellow.

Reviewer 2 Report
Comments and Suggestions for Authors
Dear Authors,
I had the pleasure of reviewing this article. The search for a replacement for CHX is certainly a topic that could prove to be very important in the future. The article itself, however, needs improvement. I will post my comments below.
Abstract:
- Undeveloped abbreviations VAS, CONSORT
Key words:
- Should be in alphabetical order and correspond to medical subject heading (mesh)
Introduction:
- Far too long introduction - maximum length of introduction should not exceed 1 page, 1/2-3/4 pages most optimal.
- At the end of the introduction should be the AIM of the study (this should not be in material and methods
- Line 104 Abbreviation PGE2 undeveloped
Material and methods:
- Line 147 - write from which year the CONSORT guideline is updated so that it is clear which one is meant.
- Inclusion and exclusion criteria should be presented in table form for clarity
- The abbreviation VAS is not developed here either
- The specified version of the ImageJ software is missing
Results:
Under each table, all abbreviations in the table should be expanded (VAS IQR)
Discussion:
The authors' method for measuring images assumes that the image was taken perpendicular to the lesion without disturbing the scale. Was there a fixed camera position relative to the lesion? If not this is a limitation that should be strongly emphasised in the discussion, as it may distort the results of the whole study with such small deviations.
Limitations in the discussion are very poorly described - please elaborate
Throughout the article (including the abstract, the number of decimal places in the presentation of significance level should be standardised)
Best regards
Reviewer
Author Response
RESPONCES REVIEWER 2
Dear Authors,
I had the pleasure of reviewing this article. The search for a replacement for CHX is certainly a topic that could prove to be very important in the future. The article itself, however, needs improvement. I will post my comments below.
- Abstract: Undeveloped abbreviations VAS, CONSORT
RE: Correction with “VAS: visual analogue scale”, “CONSORT Consolidated Standards Of Reporting Trials”
- Key words: Should be in alphabetical order and correspond to medical subject heading (mesh)
RE: Correction with “biopsy; chlorhexidine; oral cavity; pain; tocopherol; wound healing”. Exclusion of ImageJ and correction of healing.
INTRODUCTION
- Far too long introduction - maximum length of introduction should not exceed 1 page, 1/2-3/4 pages most optimal.
Re: introduction has been rewrited as requested
- At the end of the introduction should be the AIM of the study (this should not be in material and methods
RE: thanks for the clarification, corrected as requested in line 59-63
- Line 104 Abbreviation PGE2 undeveloped
RE: In rewriting the section this term is no longer present
MATHERIAL AND METHODS
- Line 147 - write from which year the CONSORT guideline is updated so that it is clear which one is meant.
RE: thank for clarification, I inserted date 2010 as follows in line 67 “(Consolidated Standards Of Reporting Trials) 2010”
- Inclusion and exclusion criteria should be presented in table form for clarity
RE: Table has been inserted as requested
- The abbreviation VAS is not developed here either
RE: Corrected with “VAS-P (Visual Analogue Scale for Pain” as reported in lines 100-101. Also corrected every VAS-D with VAS-P (D stands for “dolore”, which means “pain” in Italian translation)
- The specified version of the ImageJ software is missing
RE: Corrected inserting (ImageJ 1.53t; Java 1.8.0_345 version).
RESULTS
- Under each table, all abbreviations in the table should be expanded (VAS IQR)
RE: thanks for kindly clarification, it has been inserted as requested
DISCUSSION
- The author’s method for measuring images assumes that the image was taken perpendicular to the lesion without disturbing the scale. Was there a fixed camera position relative to the lesion? If not, this is a limitation that should be strongly emphasised in the discussion, as it may distort the results of the whole study with such small deviations.
RE: ImageJ, as described in the Materials and Methods, prevents this bias by allowing calibration based on a known measurement. As described in “Bacci C, Vanzo V, Frigo AC, Stellini E, Sbricoli L, Valente M. Topical tocopherol for treatment of reticular oral lichen planus: a randomized, double-blind, crossover study. Oral Dis. 2017 Jan;23(1):62–8.” This was assessed by measuring the length and surface area of the lesions with a gauge, taking photographs at the time of each visit, and comparing them with the ImageJ software (Wayne Rasband, US National Institute of Health, Bethesda) This program standardizes images on the basis of a measure of known length, and has been used by several authors for the purpose digital image analysis).
See also this 2007 paper: Kerner S, Etienne D, Malet J, Mora F, Monnet-Corti V, Bouchard P. Root coverage assessment: validity and reproducibility of an image analysis system. J Clin Periodontol. 2007 Nov;34(11):969-76. doi: 10.1111/j.1600-051X.2007.01137.x. Epub 2007 Sep 18. PMID: 17877749.
- Limitations in the discussion are very poorly described - please elaborate
RE: thanks for kindly clarification, limitations have been elaborated as follows in lines 375-385:
“This study has several limitations that should be considered. First, it was not possible to conduct the study in a double-blind manner due to the evident differences in consistency and application methods of the treatments (gel vs spray). Additionally, the sample size of 77 patients, while sufficient to detect primary differences, can be considered relatively small, limiting the generalizability of the results. Another limitation lies in the follow-up period, which focused on the first six postoperative days; long-term monitoring could provide more comprehensive information on the effects of the treatments on wound healing. Moreover, while height and width of the lesion were measured to assess wound healing, including the area of the lesion would have provided more robust and detailed data on the healing process. Lastly, the absence of a placebo group or a no-treatment group limits the ability to evaluate the absolute efficacy of each intervention”.
- Throughout the article (including the abstract, the number of decimal places in the presentation of significance level should be standardised)
RE: Thank you for your comment regarding the standardization of the number of decimal places for the presentation of significance levels throughout the article. We would like to clarify that the number of decimal places has been intentionally preserved to reflect the level of precision in the statistical results. For instance, reducing a value such as 0.008 to two decimal places (e.g., 0.01) would lead to a loss of precision and could misrepresent the actual significance of the result. We believe that maintaining the original precision ensures a more accurate interpretation of the data and aligns with common statistical reporting practices.
The paper has been thoroughly revised, and the changes have been highlighted in yellow.

Round 2
Reviewer 2 Report
Comments and Suggestions for Authors
Dear Authors,
You responded to all of my comments. I have no further concerns about this manuscript.
Best regards
Reviewer